# AutoNeuriteJ: An ImageJ plugin for measurement and classification of neuritic extensions

Benoit Boulan[1¤a], Anne Beghin[2], Charlotte Ravanello[1], Jean-Christophe Deloulme[1], Sylvie Gory-Fauré[1], Annie Andrieux[1], Jacques Brocard[1¤b], Eric Denarier[1]*

1 Univ. Grenoble Alpes, Inserm U1216, CEA, Grenoble Institut Neurosciences, Grenoble, France,
2 MechanoBiology Institute (MBI) at NUS Singapore MBI, T-Lab, Singapore

¤a Current address: Cellular Neurobiology Research Unit, Institut de recherches cliniques de Montréal (IRCM), Montreal, Canada
¤b Current address: PLATIM, ENS de Lyon, US8/INSERM, Lyon, France
* eric.denarier@univ-grenoble-alpes.fr

**Data Availability Statement:** All relevant data are within the manuscript and its Supporting Information files.

## Abstract

Morphometry characterization is an important procedure in describing neuronal cultures and identifying phenotypic differences. This task usually requires labor-intensive measurements and the classification of numerous neurites from large numbers of neurons in culture. To automate these measurements, we wrote AutoNeuriteJ, an imageJ/Fiji plugin that measures and classifies neurites from a very large number of neurons. We showed that AutoNeuriteJ is able to detect variations of neuritic growth induced by several compounds known to affect the neuronal growth. In these experiments measurement of more than 5000 mouse neurons per conditions was obtained within a few hours. Moreover, by analyzing mouse neurons deficient for the microtubule associated protein 6 (MAP6) and wild type neurons we illustrate that AutoNeuriteJ is capable to detect subtle phenotypic difference in axonal length. Overall the use of AutoNeuriteJ will provide rapid, unbiased and accurate measurement of neuron morphologies.

## Introduction

The study of neurodevelopmental or neurodegenerative diseases necessitates the production of mouse strains mutated in potentially disease-causing genes. To get a molecular understanding of the disease, it is classical to perform neuron cultures from these strains of mice and follow their development with the goal of revealing specific phenotypes [1]. Alternatively the effect of drugs or attractive molecules on the development of the neurons can be tested in culture [2, 3]. The early development of hippocampal neurons in culture undergoes three stages where a single round cell attaches, produces numerous growing extensions, and polarizes to differentiate one of its neurite into an axon (Fig 1) [4]. At the end of this early development, the neuron displays highly branched dendrites and axon. The description of the developmental timeline and final arborization complexity allows characterization of a morphological phenotype.

**Funding:** This work was supported by Institut National de la Santé et de la recherche Médicale, Commissariat à l'Energie Atiomique et aux Energies Alternatives, Université Grenoble Alpes and by awards from the French Agence Nationale de la Recherche to A.A. (2017-CE11-0026 MAMAs). https://www.inserm.fr/ http://www.cea.fr/ https://www.univ-grenoble-alpes.fr/ https://anr.fr/.

**Competing interests:** The authors have declared that no competing interests exist.

**Fig 1. Stages of early development of hippocampal neurons in culture.** Hippocampal neurons begin to form lamellipodia right after adhesion to the substrate (Stage 1). Few hours later the neurites sprouting begins (Stage 2). These protrusions undergo successive elongation and retraction phases with no net outgrowth. Eventually, one neurite acquires enhanced growth capabilities and becomes an axon (Stage 3).

The morphometric description of cell cultures is tedious when done by hand, confining analysis to a number of cells that may not be representative of the whole neuronal culture. Moreover, visual selection of neurons by the investigator may lead to selection bias. The use of image analysis software should overcome these problems and several plugins have been developed to help in this task. Some of plugins are neuron-based, allowing for the semi-automatic tracing of neurite extension (e.g. NeuronJ) [5, 6], or population-based, where whole neuritic lengths are measured with no neuron individualization or neuritic assignment (axon, dendrite, order) [7–9], thus losing information of the number and classification of neurites. AutoNeuriteJ was created to characterize a neuronal population in culture on a cell-based basis, with large numbers of neurons analyzed allowing an accurate estimation of the whole distribution of the population.

## Results

### AutoNeuriteJ, a plugin set to measure and classify neuritic extensions

AutoNeuriteJ has been designed to describe the neuritic arborescence of isolated neurons in cultures of different conditions or genotypes. It gives, in a text file for each neuron, the length and order (primary, secondary; etc. . .) of each neurite. It also gives measurements of the axon length, number of branches and total axonal tree length, if any. At the end of the text file a summary indicates the number of neurons that have been measured, the percentage of polarization (neurons with an axon), the mean primary neurite length (neurite that initiate form the cell body) and mean primary neurite number per neuron.

AutoNeuriteJ can be used to process a single file or multiple large images obtained from mosaic images recorded from slide scanners (Fig 2A). For a better segmentation of thin cellular processes, AutoNeuriteJ requires well-contrasted images of fluorescent cells and images of nuclei stained with DAPI or Hoechst, used to locate cell bodies and remove touching neurons.

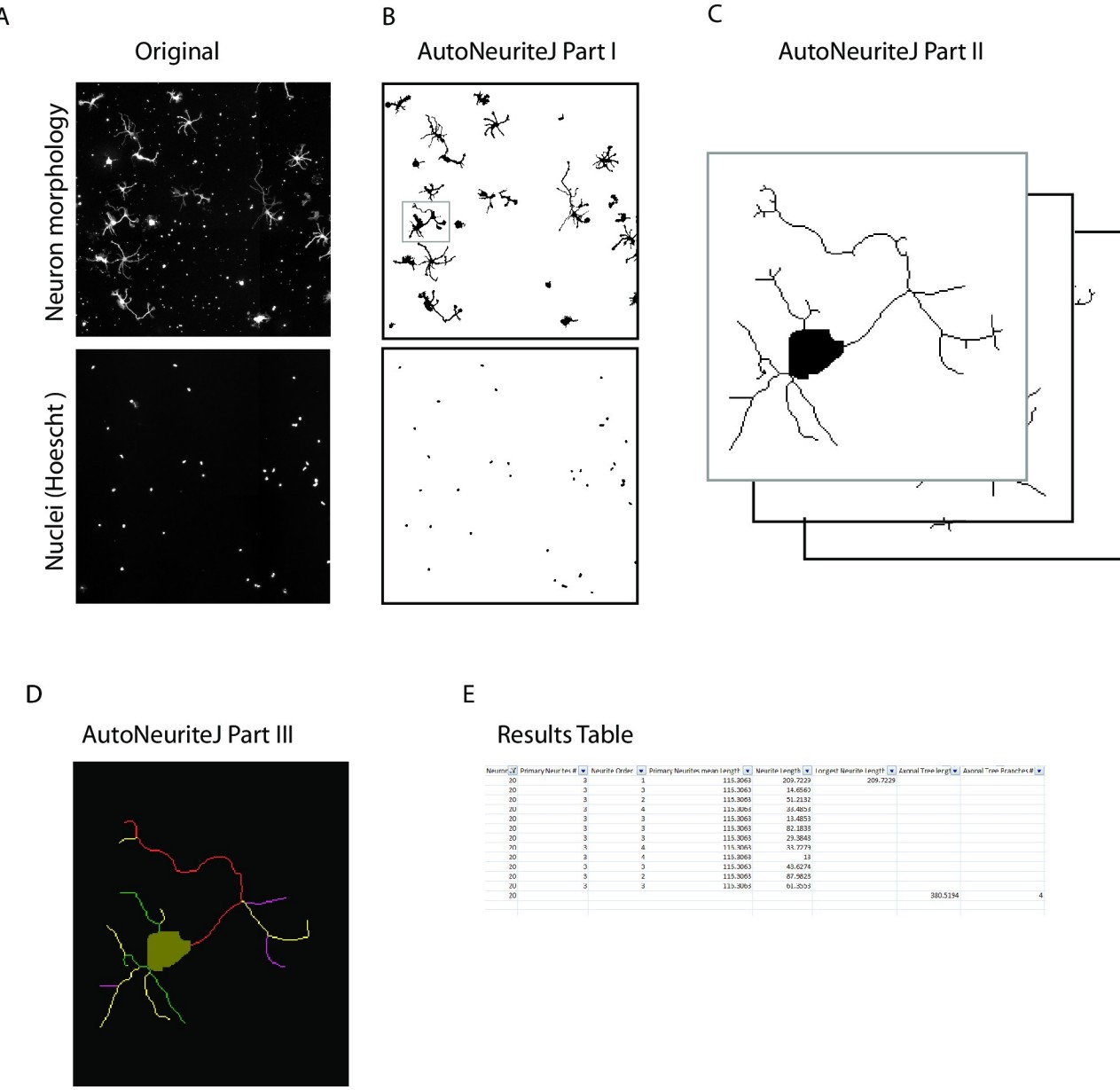

**Fig 2. Schematic of AutoNeuriteJ images processing.** (A) AutoNeuriteJ needs images of neuron (e.g. tubulin) and nuclei staining. (B) AutoNeuriteJ Part I segments nuclei and neurons, removes small particles. (C) AutoNeuriteJ part II selects individual neurons (with a single nucleus), creates cell body images and stacks of neurons. (D) AutoNeuriteJ part III detects neurite extremities, compares, classifies and measures neurites, prints results in a text file.

The whole process is divided in three independent sub-macros to allow for a better flexibility and control of different parameters defined by the users.

## Description of AutoNeuriteJ

**Part I: Segmentation of neuron and nuclei.** The first part of the macro allows the binarization of nuclei and neurons (Fig 2B). Images of neurons and nuclei are pre-processed for background homogenization and neurites enhancement. For the neuron images, a large blurred image is substracted from a median-filtered image that removes noise. For the nuclei,

a difference of Gaussians adapted to the diameter of the nuclei is applied. Parameters to be defined are the mean diameter for nuclei and the minimal surface for a binarized neuron. These parameters are used to adapt the detection of nuclei and to remove smaller particles from the images. The parameters are saved in a text file and defined as default in the subsequent use of the macro. The segmentation is supervised by the user and binarized files are saved.

**Part II: Creation of individual neuron images and skeletization.** The second part of the macro allows the creation of single neuron images from the segmented images obtained in part I (Fig 2C). It selects the cells containing a single nucleus from the binarized neurons image and produces images of single neurons and their skeleton. A cell body is also created by limited dilatation of the nucleus into the binarized neuron image. Within the macro, a minimal length for neurite and neuritic tree is set to clean short branches that could be produced by the skeletonization step and to remove undifferentiated cells without neurites. Finally, the "Region of interest" set of the selected neurons on the original image is saved. The count of neurons selected or excluded (connected neurons and undifferentiated cells) is recorded in a text file.

Note that if different neurites of the same neuron cross, it will create a loop in the skeleton that prevents measuring their length. This difficulty is overcome in AutoNeuriteJ by cutting the branch of the loop at the location of the lowest intensity of staining, using the Analyze skeleton plugin [10].

Eventually, five different stacks are created: 1) Original neurons, 2) Binarized neurons 3) Cell Bodies 4) Skeletons of the binarized neurons 5) Skeletons and cell bodies. These different stacks will be used in the next step of the process.

**Part III: Measurement and classification of neurites.** The third part of the macro is used to measure and classify neuritic extensions (Fig 2D). Using the skeleton and body of neurons generated in Part II, neurite extremities are detected for each neuron using the "binary connectivity" plugin (Gabriel Landini's Morphology Plugins).

Each neurite is at first set as "primary" and its image is created. Secondly, each neurite is compared to all other neurites to detect potential overlaps. If an overlap is detected, the order of the shortest neurite is increased (from primary to secondary etc. . .) and the longest neurite image is substracted to the shortest so that no overlaps exist. After this classification step, the length of each neurite is measured and the longest primary identified. The longest neurite will be considered as an axon if it satisfies to standard criteria [11], *i.e.* in our study the criteria were set to: the length of the axon should be twice the size of the second longest neurite and longer than 100 pixels. These parameters can be set by the user.

Finally, the macro produces an overlay for each neuron with the detected neurites colored according to their order (Fig 2D). A results text file is created presenting, for each neuron, the length and order of each neurite, the number of primary neurites, the length and number of branches of the axonal tree if any (Fig 2E). At the end of the process, a summary is printed giving the number of neuron measured, the percentage of neuron with an axon, mean and standard deviation of primary neurite length and of primary neurite number. This text file can be edited within a spreadsheet to organize the data and allow further statistical processing. Of note the stack of binarized neurons can be used to perform a classical Sholl analysis using the Fiji plugin (https://imagej.net/Sholl_Analysis).

## AutoNeuriteJ validation

The measurements given by any automatic procedure first need to be validated by a classical method used in the field of expertise. We first compared the axonal length given by

AutoNeuriteJ and when measured semi-automatically using NeuronJ. We also compared the number of primary neurites detected by eye or by Sholl analysis to AutoNeuriteJ values. Finally, we applied several treatments known to change the morphology of neurons to verify that AutoNeuriteJ can detect these changes.

**Measurement of axonal length.** The most popular ImageJ plugin for neurite measurement is NeuronJ [5]. This plugin allows a semi-automatic tracing of neurite and needs a tedious work to get large numbers of measures. We used NeuronJ to measure the axonal length of mouse neurons. On the same images of polarized neurons we applied AutoNeuriteJ and compared the techniques by correlation analysis (Fig 3A). The correlation between values obtained by NeuronJ or AutoNeuriteJ is very strong (r = 0.98). By inspecting the data corresponding to points far above the fitted line we observed that they represent cells that have not been segmented correctly (e.g. Several cells are fused and present a single nucleus). These cells can be easily removed from the binarized stacks of neurons before the neurite measurement part of AutoNeuriteJ.

**Primary neurite number.** On the same images, we also compared the primary neurite number counted by eye and detected by the macro (Fig 3B). The average values are close (2.95 and 2.91 neurite per neuron by eye and using AutoNeuriteJ, respectively) and the correlation is strong (r = 0.83), with a slight underestimation by AutoNeuriteJ due to filtering minimal neuritic length. To further validate the accuracy of AutoNeuriteJ, we compared its measurements with those of the "Sholl analysis" plugin [12]. Among the different shape descriptors proposed by the Sholl analysis plugin, we extracted the number of primary branches inferred from the count of intersections at the starting radius close to the cell body. When comparing this number to the number of primary neurites detected by AutoNeuriteJ, we found a strong correlation (r = 0.91) (Fig 3C).

From these comparisons we concluded that AutoNeuriteJ gives a reliable mean to determine axonal length and primary neurites number.

**Effect of nocodazole treatment on the major neurite length.** It has been shown that nocodazole (a drug that depolymerizes microtubule at high concentration) used at low concentration, reduces the number and the length of neurites, specifically [13, 14]. Mouse neuron cultures were treated at day 1 with 50nM nocodazole, fixed and immunostained at 3 days in vitro (DIV). After AutoNeuriteJ quantification, we found that nocodazole treatment reduces the length of the longest neurite of unpolarized cells by 25% (Fig 4A) while the axons were resistant to nocodazole treatment (non-significant reduction of 4.7% when treated). This may reflect the fact that stable MT are enriched in axons as described by Witte *et al*. [15]. We also found that Nocodazole treatment reduces the number of primary neurites in unpolarized cells (3.60 ±0.03μm when treated vs 4.50 ±0.03μm when untreated) or polarized cells (2.72 ±0.12μm, when treated vs 3.75 ±0.11μm when untreated) (Fig 4B).

From these results we concluded that, thanks to its ability to categorize the neuronal extensions, AutoNeuriteJ is a reliable tool to detect differential effects of a molecule on axons versus dendrites.

**Effect of Semaphorin 3E.** Semaphorin 3E (Sema3E) acts as a growth-promoting factor for axons of mice subicular neurons [16]. Thus we quantified Sema3E effect on subicular neurons. We first noticed the general effect of Sema3E treatment on subicular neurons polarity (10.6% of cells presenting an axon for control vs 33.9% for Sema3E treated condition). We then measure the major neuritic length. As expected, AutoNeuriteJ quantification showed an increased axonal length of polarized cells of 21.5% after 48h of treatment with Sema3E (Fig 5A). This effect is also detected on the longest neurite of unpolarized cells; with a 31.8% increase in Sema3E treated condition; revealing an early effect of Sema3E on neuritic growth. This observation at early stage can reflect either an effect on the major neurite whatever the differentiation stage of the neuron, or a global effect on all neurites of the neurons.

A

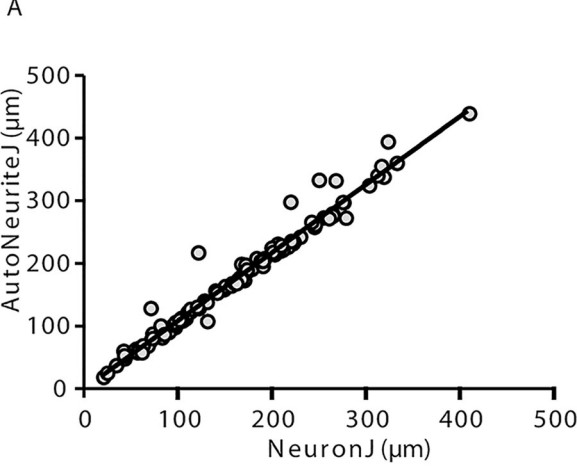

B

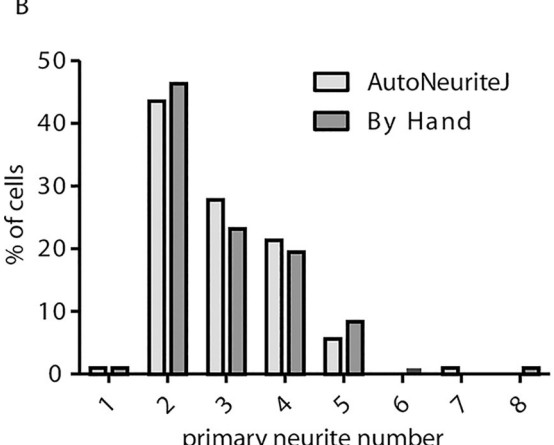

C

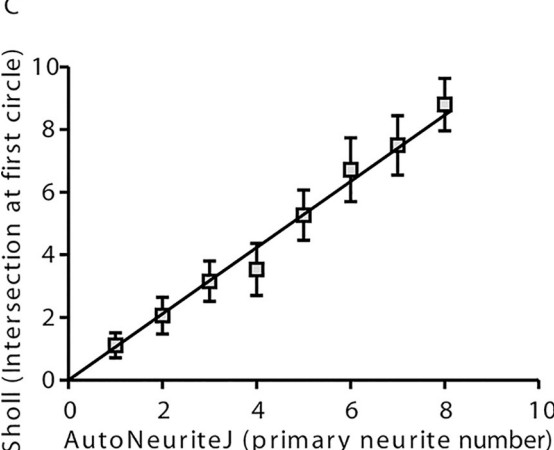

**Fig 3. Validation of axonal length and number of primary neurite measurement.** (A) Correlation of the measured axonal length using NeuronJ tracings or AutoNeuriteJ (r = 0.98 from linear regression) from 108 cells (B) Primary neurite numbers counted by hand or detected by AutoNeuriteJ (r = 0.83 from linear regression, n = 108). (C) Correlation of primary neurite number detected by AutoNeuriteJ and number of intersection at first circle detected by Sholl analysis (r = 0.91 from linear regression, n = 2062).

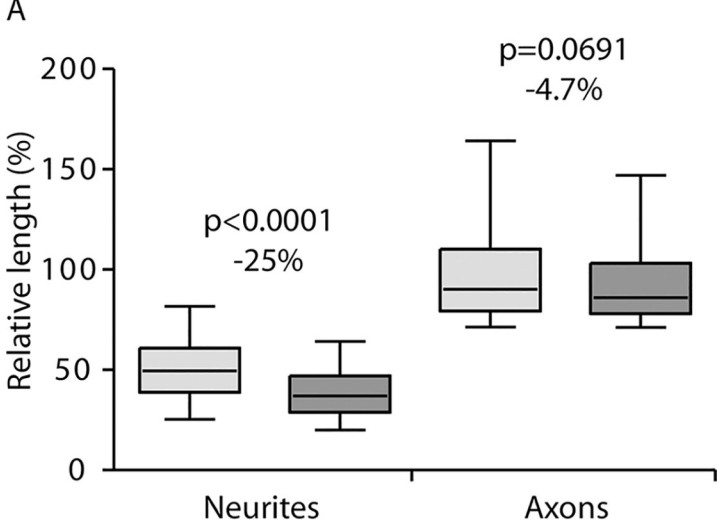

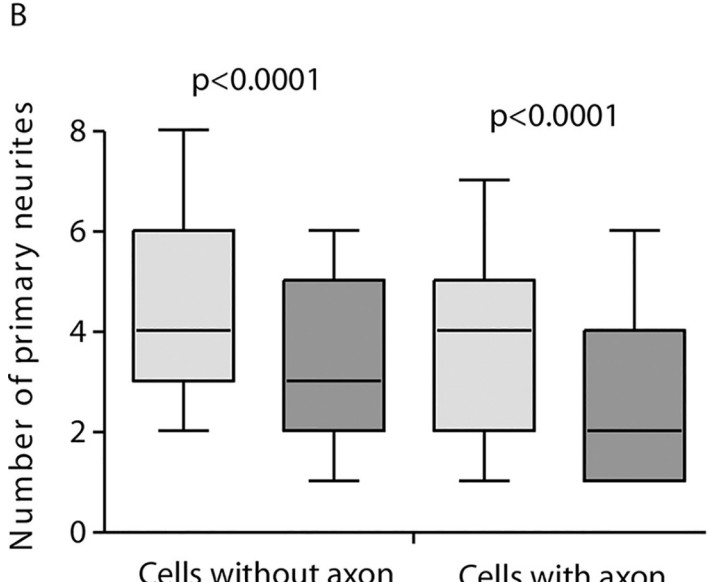

**Fig 4. Effects of nocodazole treatment on the length of major neurites and primary neurite number.** Mouse neurons in culture were treated with 50nM nocodazole 4h after plating. After 2DIV the neurons on coverslips were fixed and stained for tubulin and Hoescht before imaging and AutoNeuriteJ quantifications. (A) Nocodazole treatment reduce the size of the longest neurite by 25% (n = 3210 for control, n = 2619 for nocodazole treated) but not on axons of polarized cells (n = 297 for control and n = 175 for treated). (B) Nocodazole treatment reduces the number of primary neurites of polarized (3.70 ±0.14 vs 2.72 ±0.12 for treated) or unpolarized (4.43 ±0.14 vs 3.51 ±0.13 for treated) neurons. p values were obtained from Mann-Whitney tests.

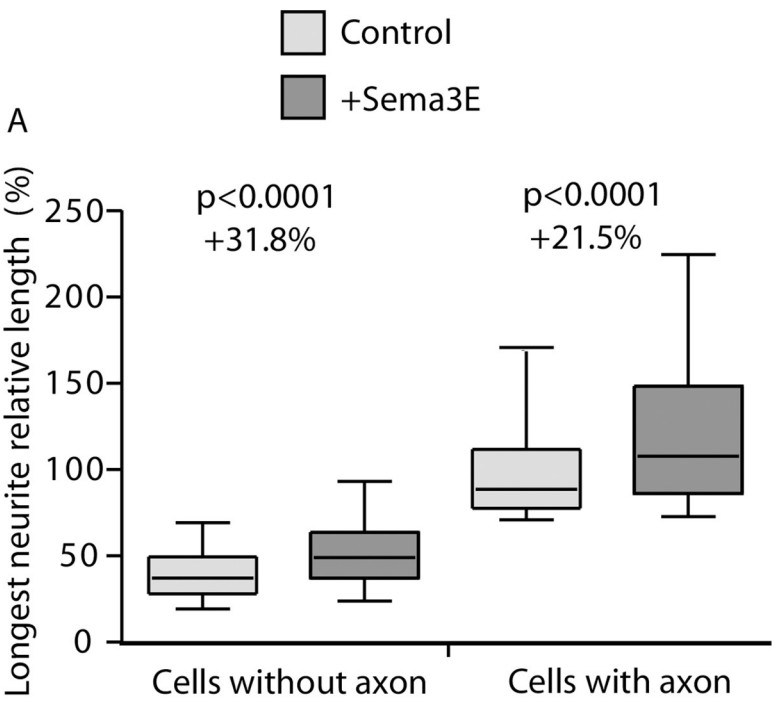

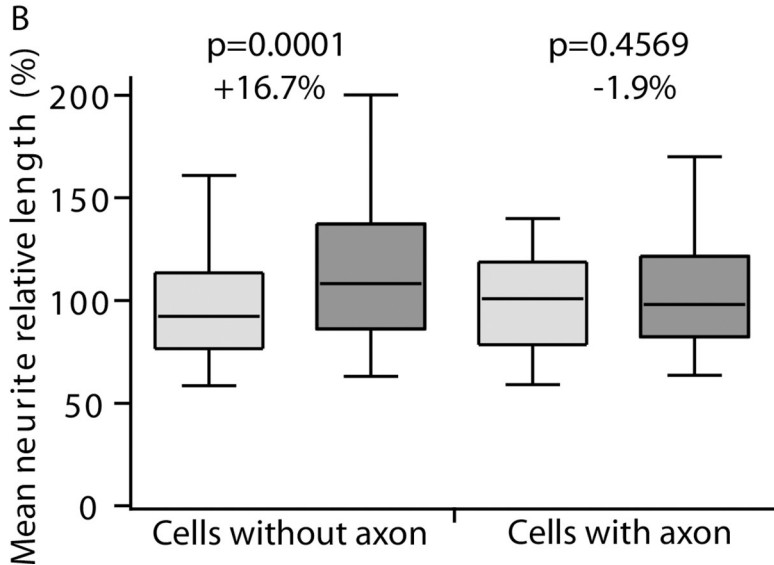

**Fig 5. Effect of Sema3E treatment on the length of major neurites.** Three independent mouse neuronal cultures were treated with Sema3E. At 2DIV, neurons were processed for quantification. (A) Length of the longest neurites for polarized and unpolarized cells treated or not with Sema3E. (Cells without axon: n = 2211 for control and n = 1512 for Sema3E treated condition; Cells with axon: n = 126 in control and n = 426 in Sema3E treated). (B) Mean length of primary neurites of polarized and unpolarized cells (Cells without axon: n = 2073 for control, n = 1473 for Sema3E treated condition; Cells with axon: n = 102 for control and n = 384 Sema3E treated). p values were obtained from Mann-Whitney tests.

Thus, we tested if the mean length of dendrites of polarized cells and the mean length of primary neurites without the longest neurite in unpolarized cells were also sensitive to Sema3E treatment (Fig 5B). As AutoNeuriteJ classify the order of neurites we collected the length of dendrites from polarized neurons (Fig 1). The length of the dendrites of polarized cells is not affected by the Sema3E treatment (Fig 5B). On the contrary, the mean length of primary neurites of unpolarized neurons excluding the longest is increased by 16.7% after Sema3E treatment (Fig 5B). We conclude that Sema3E affects all primary neurites outgrowth during early neuron developmental stages 1 and 2. During subicular neuron polarization, Sema3E induces axonal outgrowth stimulation with no effect on dendrites revealing a cell growth oriented toward the axon. The global neuritic outgrowth observed with Sema3E at early stages could be a shared property for the third class of semaphorins as neuritic outgrowth effects were also reported for Sema3A and 3C [17–19]. Nevertheless, these observations were based on global neuritic length assessments but no direct comparison between axonal versus dendritic effects of semaphorins had been reported until now.

Overall, these results showed that AutoNeuriteJ easily allows a direct comparison of several parameters of neuronal arborization at different developmental stages.

## Morphological analysis of MAP6 knock-out neurons

In our lab, we produced a mouse MAP6 knockout. MAP6 encodes for a protein that stabilized microtubules and is present in their lumen [20]. The MAP6 KO mice present numerous behavioral defects that were alleviated by neuroleptic treatment. Therefore they are a model of psychiatric disorders such as schizophrenia or depression [21, 22]. These mice also present brain anatomy perturbations where white matter tracts are altered as in fornix, corpus callosum or pyramidal tracts [2, 23]. The analysis of neurons in culture from these mice have shown that they present a defect in the Sema3E response and perturbation of the actin filament organization in dendritic spines [2, 24]. No detailed morphometric analysis of these neurons has been done yet. Taking advantages of automatic quantification, we cultivated hippocampal neurons from MAP6 KO or WT mouse littermates. The coverslips were recovered after 48h in culture, stained with antibody against tubulin and neuronal morphology analyzed with AutoNeuriteJ (Fig 6). The polarization index was increased in MAP6 KO neuron cultures (from 9.2% for WT neurons to 16.1% for MAP KO). Moreover, we were able to detect that MAP6 KO neurons produced longer axons than the controls by 23.5% (Fig 6A). Furthermore, thanks to the ability of AutoNeuriteJ to identify axonal primary and secondary branches we performed a measurement of the total axonal tree (axon plus its branches) in both genotypes. We found an increase of 20.2% of the size of MAP6 KO axonal tree as compared to WT neurons (Fig 6B). AutoNeuriteJ also allows determining the mean internode distance (axonal tree divided by the number of branches) which was also found increased by 19.7% in MAP6 KO neurons as compared to WT (Fig 6C). Thus MAP6 KO neurons increase all their axonal branches to the same extend after specification of the axon resulting in an increased internodes length. Overall, MAP6 KO experiments show the capacity of AutoneuriteJ to detect variations of both classical and more subtle morphological features of neurons in culture.

## Discussion

We developed a new ImageJ macro able to measure neuron morphology in several neuronal culture conditions. Existing plugins provide only global measurement of neuronal growth, while AutoNeuriteJ allows for quantification of individual neurons. AutoNeuriteJ can also give

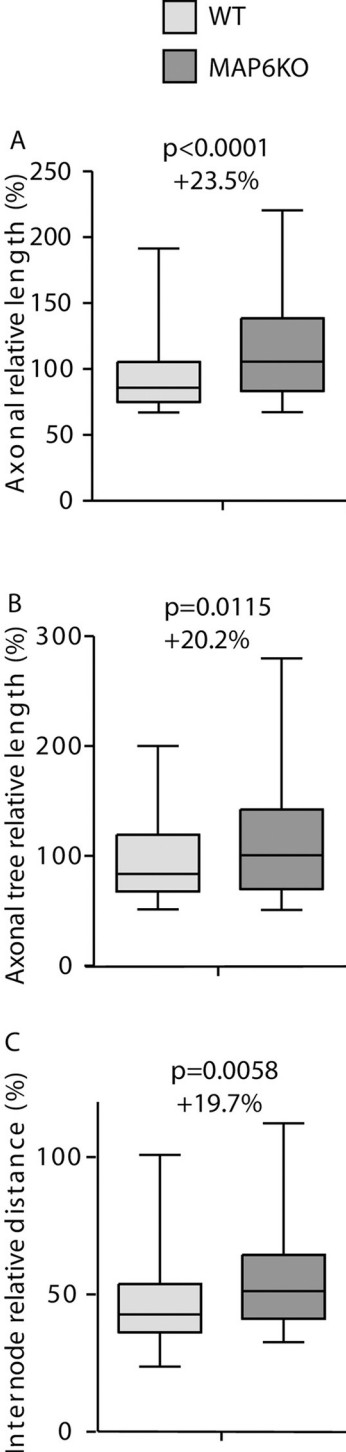

**Fig 6. MAP6 deletion enhances axonal growth.** Neurons from control and MAP6 KO mice from three independent experiments were processed for quantification. (A) Axon length for WT and MAP6 KO mice (n = 120 for WT, n = 207 for MAP6 KO). (B) Axonal tree length of polarized neurons (n = 120 for WT, n = 207 for MAP6 KO). (C) Internode distance of polarized neurons (n = 54 for WT, n = 117 for MAP6 KO). p values were obtained from Mann-Whitney tests.

a global analysis of the culture by building a stack of the segmented neurons with specific colors for the neurites order, overlaid on the original image.

The macro produces a neuron-centric analysis with the classification of neurite. This provides the possibility to analyze crucial steps during neuronal differentiation and maturation. The classification of neurites gives the percentage of polarization of the culture, an essential marker of neuronal differentiation. It also allows the distinction of the effect of a molecule on different types of neurites as demonstrated in this study with the differential effect of nocodazole on axons and dendrites. To our knowledge, without AutoNeuriteJ, this was only achievable by manual tracing.

We performed Sema3E experiment as a validation test of AutoNeuriteJ. We observed, as expected, an increased length of the major neurites in polarized and unpolarized neurons. In polarized neuron, we found that the dendrites do not respond anymore to Sema3E treatment. Interestingly CRMP2, a downstream effectors of semaphorins, is enriched in axons whereas it is uniformly expressed in all neurites in unpolarized neurons [25]. We can thus speculate that the relocalization of CRMPs toward axons could limit the dendritic response to Sema3E stimulation after axonal specification.

AutoNeuriteJ also detects various morphologic differences between WT and genetically modified neurons allowing proposing mechanistic scenarios. The experiments on MAP6 KO neurons show that, consistent with its role of stabilization of microtubule, the absence of MAP6 increases the dynamics of microtubule and allows a faster growth of the axon and of the axonal tree. These results strengthen the previously described role of MAP6 in axonal development [26].

In conclusion, AutoNeuriteJ is an accessible tool that facilitates the classification and measurement of neurites. Hopefully it will make easier, for other investigators, the morphometric analysis of neurons during early differentiation.

## Material & methods

### Hippocampal and subicular neuron cultures

In accordance with the policy of the Institut des Neurosciences of Grenoble (GIN) and the French legislation, experiments were done in compliance with the European Community Council Directive of 24th November 1986 (86/609/EEC). The research involving animals was authorized by the Direction Départementale de la protection des populations—Préfecture de l'Isère-France and by the ethics committee of GIN number 004 accredited by the French Ministry of Research.

Brains from embryos (E17.5) were dissected and hippocampus or subiculum were removed. Selected parts were dissociated and plated onto polylysine/laminin-coated as previously described [2]. The axonal growth of subicular neurons were stimulated by addition of control AP or Sema3E-AP (6.3nM) in supernatant 2h after cells platting [2]. After 2DIV, cultured cells were fixed and immunostained. Hippocampal neurons were treated by addition of 50nM nocodazole in the medium. After 2DIV, cultured cells were fixed and immunostained.

### Immunostaining

Cells were fixed at 37°C in 4% paraformaldehyde, 4% sucrose in phosphate buffered saline (PBS) and permeabilized using 0.1% Triton X-100 in PBS 1min. Cells were then incubated with α3A1 mouse primary antibody produced in the laboratory (1:10,000) for 1h at room temperature. Cells were washed thrice in PBS 0.1 tween and incubated with cyanine-3 conjugated secondary antibody (1:1000) for 1h at room temperature. After three final washes, nuclei were stained using Hoechst 33258 (1 μg/ml) in the mounting media (Dako).

## Image acquisition

For manual neuron morphometry, images of each neuron were acquired using 20 X N.A 0.5 objective on an Axioskop 50 microscope (Zeiss). All images were combined in a single file for an easier manipulation. Measure of axonal length was performed using NeuronJ [5].

For automated neuron morphometry, mosaic images of 2 or more ROI per condition were acquired using a 20x N.A 0.8 objective on an Axioscan Z1 (ZEISS) microscope. Mosaic images were then processed with AutoNeuriteJ.

## Statistical analysis

Statistical analyses were performed using Prism 5 (GraphPad). For the Sema3E and MAP6 KO experiments, values were normalized for each independent experiment to the value of the mean axonal length in the corresponding control condition. Weighting of independent experiments was obtained by random draw of equal number of neurons in each experiment before pooling values for statistical analysis. In box and whiskers figures the whiskers represent 5 to 95% of the values. Percentage of variation indicated in Figs 5 and 6 are calculated from the median values. For all figures the statistical tests used are indicated in the figure legends.

## User recommendations and drawbacks

**Culture density and immunofluorescence.** Several antibodies can be used for AutoNeuriteJ analysis. Our general recommendation here would be to use markers widespread in all neurons with a strong signal in order to accurately detect smaller neurite structures by setting low detection thresholds. As tubulin is the major protein of neurons, several anti-tubulin antibodies are useful. In the present study, we used a homemade anti-alpha tubulin antibody (α3A1) but other anti tubulin can be used as a classical anti-βIII tubulin specific for neuronal cells or anti-acetylated tubulin antibody that detects only stable microtubules, present in the shafts of neuritic extensions.

AutoNeuriteJ was developed to analyze the first steps of neuron differentiation. It does not consider neurons with overlapping neurites, thus creating a bias in the population considered. This problem also occurs when the measures are done by hand, it is generally circumvented in the field by using low density culture of neurons. To this end it is very important for users to verify a low neuronal density before performing immunostaining.

**Image resolution.** As AutoneuriteJ is dedicated to neurite measurement, it does not need high resolution images to properly operate (1μm pixel size images are recommended). High resolution images will increase the duration of image processing as neurite measurement is based on pixel erosion. To circumvent this issue, AutoNeuriteJ part I performs scaling of images.

**Quantification speed and computer requirements.** Time for completion of this macro depends mostly on the number of counted neurons and neurites. From our experience we recommend to analyze images with no more than 500 neurons. This number is sufficient for comparing several groups of treatment or genotypes. If more neurons are needed, it is possible to run several occurrences of Fiji in parallel. As an example, we used eight occurrences of Fiji running at the same time on the same computer to measure 30,000 neurons in less than 12h.

## Availability

AutoNeuriteJ and a user guide are available at GitHub (https://github.com/Grenoble-Institute-Neurosciences/AutoNeuriteJ). First and last authors may also be contacted for details or future collaborations.

## Supporting information

**S1 Data.**
(ZIP)

**S2 Data.**
(ZIP)

## Acknowledgments

The authors want to thank Thomas Brown for his comments and manuscript corrections. We also thank S. Andrieu, M. Lapierre, L. Romian, F. Mehr, and F. Rimet for animal care and F. Vossier and L. De Macedo for mouse genotyping. The Photonic Imaging Center of Grenoble Institute Neuroscience (Univ Grenoble Alpes–Inserm U1216) is part of the ISdV core facility and certified by the IBiSA label.

## Author Contributions

**Conceptualization:** Benoit Boulan, Anne Beghin, Eric Denarier.

**Formal analysis:** Jacques Brocard.

**Funding acquisition:** Annie Andrieux.

**Investigation:** Charlotte Ravanello, Jean-Christophe Deloulme, Sylvie Gory-Fauré.

**Methodology:** Jacques Brocard.

**Resources:** Sylvie Gory-Fauré.

**Software:** Benoit Boulan, Anne Beghin, Eric Denarier.

**Writing – original draft:** Benoit Boulan, Eric Denarier.

**Writing – review & editing:** Benoit Boulan, Annie Andrieux, Eric Denarier.

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
