## [Decision Letter · Decision Letter 0]

9 Jun 2020

PONE-D-20-15371

AutoNeuriteJ: An ImageJ plugin for measurement and classification of neuritic extensions

PLOS ONE

Dear Dr. Denarier,

Thank you for submitting your manuscript to PLOS ONE. After careful consideration, we feel that it has merit but does not fully meet PLOS ONE’s publication criteria as it currently stands. Therefore, we invite you to submit a revised version of the manuscript that addresses the points raised during the review process (see below).

We look forward to receiving your revised manuscript.

Kind regards,

Giorgio F Gilestro, PhD

Academic Editor

PLOS ONE

Journal Requirements:

Reviewers' comments:

Reviewer's Responses to Questions

**Comments to the Author**

1. Is the manuscript technically sound, and do the data support the conclusions?

Reviewer #1: Yes

2. Has the statistical analysis been performed appropriately and rigorously? 

Reviewer #1: Yes

3. Have the authors made all data underlying the findings in their manuscript fully available?

Reviewer #1: Yes

4. Is the manuscript presented in an intelligible fashion and written in standard English?

Reviewer #1: Yes

5. Review Comments to the Author

Reviewer #1: This manuscript describes AutoNeuriteJ, a new ImageJ plugin for measuring the length and classifying neurites from fluorescently labelled neurons. The authors validate this plugin by benchmarking against manual annotation, and by analysis of neurite length under experimental perturbations.

This work meets all the criteria for publication as listed in the PLOS ONE site. The results of the validation experiments support the conclusion that AutoNeuriteJ is a reliable and useful tool for analyzing neurite morphology that could see adoption by many labs.

I have minor comments and suggestions:

1. Please briefly describe the criteria for indicating if a neuron possesses an axon/is polarized. This information could be presented in Part III: Measurement and classification of neurites.

2. In Figure 3A, the biggest outliers are all above the line, indicating that when analyzed by AutoNeuriteJ, these outliers have ~10-70% greater lengths as estimated by eye from Figure 3A This result suggests that the outliers are not random but may be due to some systematic difference in the measurement process for about 5% of the neurons. This is not a serious issue but if the authors have some insights into how these discrepancies could arise, it would be very useful to discuss this briefly since it can help users understand how to optimize their analysis.

3. There are some minor typos and phrases that could be tweaked for better clarity:

- Figure 5 legend: "Length of the longest neurites for cells presenting an axon or not treated with Sema3E" should be rephrased or have a comma after "not" to remove ambiguity and make clear that the groups are not "presenting an axon" and "not treated with Sema3E."

- Page 8, paragraph 2: "the mean length of primary neurites without the longest in unpolarized cells" - this phrase needs a noun after the word "longest."

- Page 12, paragraph 1: "wiskers" is a typo - should be "whiskers"

6. PLOS authors have the option to publish the peer review history of their article (what does this mean?). If published, this will include your full peer review and any attached files.

Reviewer #1: No

---

## [Author Response · Author response to Decision Letter 0]

11 Jun 2020

Dear Editor,

We thank the reviewer and the editor for their interest in our work.

We corrected the manuscript according to the reviewer's recommendations. We hope that our manuscript is now acceptable for publication in Plos One.

Response to the reviewer :

"1. Please briefly describe the criteria for indicating if a neuron possesses an axon/is polarized. This information could be presented in Part III: Measurement and classification of neurites"

We clarified the criteria used in our study in the text :

The longest neurite will be considered as an axon if it satisfies to standard criteria [11], i.e. in our study the criteria were set to : the length of the axon should be twice the size of the second longest neurite and longer than 100 pixels . These parameters can be set by the user.

"2. In Figure 3A, the biggest outliers are all above the line, indicating that when analyzed by AutoNeuriteJ, these outliers have ~10-70% greater lengths as estimated by eye from Figure 3A This result suggests that the outliers are not random but may be due to some systematic difference in the measurement process for about 5% of the neurons. This is not a serious issue but if the authors have some insights into how these discrepancies could arise, it would be very useful to discuss this briefly since it can help users understand how to optimize their analysis."

We obtained insight for the outliers and added theses informations in the text.

By inspecting the data corresponding to points far above the fitted line we observed that they represent cells that have not been segmented correctly (e.g. Several cells are fused and present a single nucleus). These cells can be easily removed from the binarized stacks of neurons before the neurite measurement part of AutoNeuriteJ.

"3. There are some minor typos and phrases that could be tweaked for better clarity:- Figure 5 legend: "Length of the longest neurites for cells presenting an axon or not treated with Sema3E" should be rephrased or have a comma after "not" to remove ambiguity and make clear that the groups are not "presenting an axon" and "not treated with Sema3E."

The sentence has been rewritten to : (A) Length of the longest neurites for polarized and unpolarized cells treated or not with Sema3E

"4- Page 8, paragraph 2: "the mean length of primary neurites without the longest in unpolarized cells" - this phrase needs a noun after the word "longest."- Page 12, paragraph 1: "wiskers" is a typo - should be "whiskers"

The manuscript has been corrected accordingly.

---

## [Editor Report · Decision Letter 1]

30 Jun 2020

AutoNeuriteJ: An ImageJ plugin for measurement and classification of neuritic extensions

PONE-D-20-15371R1

Dear Dr. Denarier,

We’re pleased to inform you that your manuscript has been judged scientifically suitable for publication and will be formally accepted for publication once it meets all outstanding technical requirements.

Kind regards,

Giorgio F Gilestro, PhD

Academic Editor

PLOS ONE
---

## [Editor Report · Acceptance letter]

6 Jul 2020

PONE-D-20-15371R1 

AutoNeuriteJ: An ImageJ plugin for measurement and classification of neuritic extensions 

Dear Dr. Denarier:

I'm pleased to inform you that your manuscript has been deemed suitable for publication in PLOS ONE. Congratulations! Your manuscript is now with our production department. 

Kind regards, 

on behalf of

Dr. Giorgio F Gilestro 

Academic Editor

PLOS ONE